# The Influence of Air Pollutants and Meteorological Conditions on the Hospitalization for Respiratory Diseases in Shenzhen City, China

**DOI:** 10.3390/ijerph18105120

**Published:** 2021-05-12

**Authors:** Shi Liang, Chong Sun, Chanfang Liu, Lili Jiang, Yingjia Xie, Shaohong Yan, Zhenyu Jiang, Qingwen Qi, An Zhang

**Affiliations:** 1National Key Clinical Specialty of Occupational Diseases, Shenzhen Occupational Diseases Prevention and Treatment Center, Shenzhen 518020, China; carlsl@126.com (S.L.); sunchong2093@gmail.com (C.S.); xingzhe045@163.com (Z.J.); 2Shenzhen Environmental Monitoring Center, Shenzhen 518000, China; chanfangliu@meeb.sz.gov.cn (C.L.); xieyingjia@meeb.sz.gov.cn (Y.X.); yanshaohong@meeb.sz.gov.cn (S.Y.); 3National Key Laboratory of Resources and Environmental Information System, Institute of Geographic Sciences and Natural Resources Research, Chinese Academy of Sciences, Beijing 100101, China; jiangll@igsnrr.ac.cn

**Keywords:** air pollutants, meteorology, respiratory diseases, distributed non-linear models

## Abstract

Air pollutants have significant direct and indirect adverse effects on public health. To explore the relationship between air pollutants and meteorological conditions on the hospitalization for respiratory diseases, we collected a whole year of daily major air pollutants’ concentrations from Shenzhen city in 2013, including Particulate Matter (PM_10_, PM_2.5_), Nitrogen dioxide (NO_2_), Ozone (O_3_), Sulphur dioxide (SO_2_), and Carbon monoxide (CO). Meanwhile, we also gained meteorological data. This study collected 109,927 patients cases with diseases of the respiratory system from 98 hospitals. We investigated the influence of meteorological factors on air pollution by Spearman correlation analysis. Then, we tested the short-term correlation between significant air pollutants and respiratory diseases’ hospitalization by Distributed Lag Non-linear Model (DLNM). There was a significant negative correlation between the north wind and NO_2_ and a significant negative correlation between the south wind and six pollutants. Except for CO, other air pollutants were significantly correlated with the number of hospitalized patients during the lag period. Most of the pollutants reached maximum Relative Risk (RR) with a lag of five days. When the time lag was five days, the annual average of PM_10_, PM_2.5_, SO_2_, NO_2_, and O_3_ increased by 10%, and the risk of hospitalization for the respiratory system increased by 0.29%, 0.23%, 0.22%, 0.25%, and 0.22%, respectively. All the pollutants except CO impact the respiratory system’s hospitalization in a short period, and PM10 has the most significant impact. The results are helpful for pollution control from a public health perspective.

## 1. Introduction

Fossil fuels have become the blood of civilization. The global total oil consumption per day increased from 85,665 to 100,959 thousand barrels from 1999 to 2019 [1]. While enjoying the advancements of civilization, human beings have to face severe impacts of air pollution. In recent years, many epidemiological studies have shown that significant air pollutants such as particulate matter, nitrogen dioxide, sulfur dioxide, and ozone have significant direct and indirect adverse effects on public health [2,3,4,5,6]. According to “the global burden of disease study,” only PM_2.5_, one of the air pollutants, led to 2.94 million all-cause deaths and 83 million disabled-adjusted life-years global in 2017. It ranks 10th among the global risk factors for death [7]. Research shows that SO_2_ is associated with CVD risk (cardiovascular disease), the risk of death from respiratory disease, and total death [8,9,10]. There have been many studies on air pollution carried out in China in recent years, mainly focusing on cities that are very densely populated but also relatively severely polluted, such as Beijing [11,12], Shanghai [13], Chongqing [14], and Shenyang [15]. Studies show that complicated factors influence the health effects of air pollution. For example, PM_2.5_ components from different sources have significant differences [16] and divergent impacts on pathogenic microorganisms [17] through complex interactions and transformations following temperature, humidity, and other factors. These complex factors are challenging to investigate and control accurately in current epidemiological studies. Therefore, it is not reliable to determine the effects of air pollution in one area based on studies in other areas on health [18].

Shenzhen is a Special Economic Zone city in China, which has a High urbanization level, highly developed economy, and high-density traffic [19]. Its air quality is better than other similar cities in China. Under the model of high development-low pollution, Shenzhen city offers a valuable opportunity to study the effects of air pollution on health. Whether there is an exposure time or accumulation of air pollutants affecting public health is also an important issue in environmental health research [20].

This study aimed to investigate the effects of air pollutants and meteorological conditions on residents’ hospitalization for respiratory diseases in the short term and compare different pollutants’ exposure time or accumulation effects. This study’s air pollutants include Particulate Matter (PM_10_, PM_2.5_), Nitrogen dioxide (NO_2_), Ozone (O_3_), Sulphur dioxide (SO_2_), Carbon monoxide (CO). The meteorological factors include temperature, humidity, and wind direction.

## 2. Materials and Methods

### 2.1. The Information about Shenzhen City

Shenzhen city is located in the south of China, which links Hong Kong SAR, China, through the Shenzhen River. Shenzhen has a Marine subtropical climate. The city has ten districts, with an area of 1997 square kilometers and a resident population of 13 million. It has a mild climate with an annual average temperature of 22.4 °C and abundant rainfall. Its rainy season lasts from April to September every year, with an annual rainfall of 1933 mm and an average annual sunshine duration of 2120 h. The prevailing wind direction is from east–southeast all year round, but sometimes the wind direction is from north-northwest in winter. By 2019, the number of vehicles in Shenzhen was 3.5 million, but the number of new-energy vehicles ranked first in China.

### 2.2. Air Quality Data and Meteorological Data

To match the time of hospital admission data, we obtained air quality data from the Shenzhen environmental monitoring center, including the daily air pollutant mass concentration at each station from January 1st to 31 December in 2013 (data missing on January 31th). There were 19 air quality monitoring stations in Shenzhen, which monitored major air pollutants’ concentrations (PM_10_, PM_2.5_, NO_2_, SO_2_, O_3_, CO) in real-time (Figure 1). The simulation process for pollutant dispersion models requires multiple environment variables simultaneously in the atmospheric transport’s complex physical and chemical processes. Moreover, the data acquisition of these multiple environment variables is difficult and costly. Thus, the average daily concentration of each air pollutant as the exposure concentration was calculated based on the kriging interpolation method in each 1 km grid. Meanwhile, we gathered daily average weather conditions meteorological data (temperature, humidity, and wind direction) of Shenzhen from the open website http://lishi.tianqi.com/shenzhen/2013 (accessed on 1 June 2018).

### 2.3. Hospital Admission Data

Due to the medical information data release policy in Shenzhen, we cannot access the latest data. The data of 129,319 inpatients with ICD-10 code j00-j99 (Diseases of the respiratory system) from 98 hospitals of Shenzhen city was gained from Shenzhen medical information center from 1 January 2013 to 31 December 2013. The data included the primary diagnosis, admission date, discharge date, age, gender, and residential location. We used a total of 109,927 patients cases in the study. We excluded 19,052 patients with non-local addresses and 340 patients with pulmonary diseases caused by external causes (ICD-10 code: j60-j70). Personal air pollution exposure does change with the movement of a person’s spatial location, but we do not have these population movement data. Because we only have the residential location, we geocode the residential location to match exposure concentration in each 1 km grid.

### 2.4. Methods

We adopted Spearman correlation analysis to analyze the relationship between the air pollutants concentration. In the correlation analysis of atmospheric pollutant concentration and meteorological factors, we included the average daily concentration of six Air pollutants (the unit of CO is mg/m^3^, the unit of the rest is μg/m^3^), Air Quality Index (AQI), average daily temperature, humidity, and wind direction. We did the following processing for wind speed and direction: we regarded 0–3 wind force as no wind and recorded wind force of 4 or above. The original eight wind directions were integrated into four wind directions by vector calculation method, and the data finally included in the analysis was the wind force value of each wind direction. We selected Spearman correlation analysis based on RANK to ensure the research results’ stability, considering the non-normal distribution of air pollutants and the possible outlier, maximum, and minimum value.

This paper used the Distributed Lag Non-linear Model (DLNM) package in R (Version 3.5.1, University of Auckland, Auckland, New Zealand) to analyze the short-term association between significant air pollutants (SO_2_, NO_2_, PM_10_, PM_2.5_, O_3_, CO) and meteorological conditions on residents’ hospitalization for respiratory diseases. The dependent variable was the number of daily hospital admissions, and the primary independent variable was the daily concentration of pollutants and meteorological conditions.

The equation was of the form:$$\;logE\left[Y_{t}\right]=\alpha +\beta_{1}X_{t,l}+\beta_{2}T_{t,l}+NStime，7+NShum,3+DOW$$
where $Y_{t}$ represents the number of hospitalized patients with respiratory diseases on day *t*, α is the intercept, $\beta_{1}$ and $\beta_{2}$ are the parameter vectors, $X_{t,l}$ is the cross basis matrix of pollutant concentration, $T_{t,l}$ is the cross basis matrix of air temperature, *l* is the lag days, NS is the natural spline function, *time* is the date variable, *hum* is the humidity variable, Dow is the week variable.

It is worth noting that the unit of pollutant concentration change of the models in most studies is 10 μg/m^3^, making it obscure and abstract to compare the effects on the health of different pollutants. Because of the wide differences in baseline concentrations of different pollutants, it is too wide and hard to intuitively compare the estimation of the health effects obtained by using 10 μg/m^3^ as the same unit makes. Considering this problem, we use 10% of each pollutant’s annual average concentration as the unit of change in concentration in the model. Besides, the influence of temperature and humidity on the respiratory system cannot be ignored [20,21]. Therefore, we used the natural cubic spline function of 4 degrees of freedom to adjust the influence of temperature and relative humidity. Finally, we added a dummy variable for the day of the week.

## 3. Results

### 3.1. Descriptive Statistics

Table 1 summarizes the descriptive statistics of air pollution, meteorological factors, and the number of hospital admissions in Shenzhen during the study period. Figure 2 shows the air pollutant data in a time series analysis model. The average concentration of PM_2.5_ is 40.22 μg/m^3^, and the maximum value is 135.81 μg/m^3^, which is higher than both the air quality standard of China (annual average is 35 μg/m^3^, and the daily average is 75 μg/m^3^, and World Health Organization (WHO) air quality standard (annual average is 10 μg/m^3^ and the daily average is 25 μg/m^3^). The number of days when the concentration of PM_2.5_ in Shenzhen exceeds the China standard is 38 days, and 237 days according to WHO standard during the study period. The average concentration of PM_10_ is 61.3 μg/m^3^, and the maximum value is 184.78 μg/m^3^. According to China’s standard, the annual average and the daily average is 70 μg/m^3^ and 150 μg/m^3,^ respectively, while for WHO’ air quality standard, the annual average is 20 μg/m^3^, and the daily average is 50 μg/m^3^ respectively. For PM_10_ of Shenzhen, the number of days is five days above the China standard and 176 days above the WHO standard. The average concentration of other air pollutants’ situation is lower than or equal to the WHO and China standards. During the study period, the average temperature was 23.85 °C, and the average humidity was 75.62%. The average daily number of inpatients was 307.92, and there were significant differences between males and females.

### 3.2. The Relationship between the Concentration of Air Pollutants, Meteorological Factors, and Wind Direction

Table 2 shows the relationship between the concentration of air pollutants, meteorological factors, and wind direction. The average daily concentration of the six air pollutants showed a significant positive correlation with each other (*p* < 0.01). The correlation between PM_10_ and PM_2.5_ was the strongest (r = 0.947), followed by SO_2_ and PM_10_ (r = 0.831), and then SO_2_ and PM_2.5_ (r = 0.815). Among the six pollutants, the correlation between particulate matter (PM_10_, PM_2.5_) and various pollutants other than CO was relatively strong, while the correlation between CO and other pollutants was relatively weak. We calculated AQI (air quality index) by the maximum IAQI (air quality index of pollutants). Therefore, the correlation between AQI and six pollutants reflects the contribution of 6 pollutants to AQI in Shenzhen to some extent. The correlation between the two kinds of particles (PM_10_, PM_2.5_) and AQI was the strongest, followed by SO_2_ and NO_2_, and O_3_ and CO did not reach the level of strong correlation with AQI (r> and 0.7). Air temperature, humidity, and six air pollutants were negatively correlated. There was a significant negative correlation between the north wind and NO_2_, while there was a weak positive correlation between the north wind and the other five pollutants. There was a significant negative correlation between the south wind and the six pollutants, and the correlation was generally stronger than that of the north wind. There is a significant negative correlation between the east wind and SO_2_, NO_2_, but not with the other four pollutants. There is a significant negative correlation between the west wind and NO_2_, but not with the other five pollutants.

### 3.3. The Relationship between Six Air Pollutants and the Whole Number of Respiratory Inpatients

Figure 3 shows the relationship between six air pollutants and the whole number of respiratory inpatients. Except for CO, there is a significant positive correlation between other air pollutants and the number of respiratory inpatients during the lag period. RR peak of PM_10_ is lag0. RR peak of PM_2.5_, SO_2_, NO_2_, and O_3_ is lag5. At a lag of 0 days, PM_10_ increased by an annual average of 10%, and the risk of respiratory hospitalization increased by 0.4% (RR, 1.003976; 95%CI, 1.000001–1.007967). When the time lag was 5 days, PM_10_, PM_2.5_, SO_2_, NO_2_, O_3_ increased by an annual average of 10%, and the risk of respiratory system hospitalization increased by 0.29% respectively (RR, 1.002923; 95%CI, 1.000966–1.004883), 0.23% (RR, 1.002261; 95%CI, 1.000461–1.004065), 0.22% (RR, 1.002174; 95%CI, 1.000048–1.004305), 0.25% (RR, 1.002514; 95%CI, 1.000321–1.004712), 0.22% (RR, 1.002173); 95%CI, 1.000402–1.003947).

Figure 4 shows each pollutant’s impact on the number of hospitalizations for respiratory diseases in different populations. CO had no significant effect on hospitalization risk in any population. RR peak concentrated in lag0-lag2 in the female group exposed to PM_10_, PM_2.5_, SO_2_, and NO_2_. Moreover, it was earlier than that in the male group exposed to PM_10_, PM_2.5_, SO_2_, and NO_2_. Exposure to O_3_ did not significantly impact hospitalization risk in the male group, while the RR peak in the female group occurred with a lag of 5 days, which was different from the other pollutants. From the perspective of age, the older (age ≥65) group’s hospitalization risk exposed to all air pollutants was higher than that of other age groups, and the RR peak concentrated in lag4–lag6.

## 4. Discussion

There was a significant negative correlation between the north wind and NO_2_, while there was a weak positive correlation between the north wind and the other five pollutants. Although it had a significant negative correlation between the south wind and six pollutants, the correlation was generally more reliable than the north wind. There is a significant negative correlation between the east wind and SO_2_, NO_2_, but not with the other four pollutants. There is a significant negative correlation between the west wind and NO_2_, but not with the other five pollutants.

As is shown in Figure 5, this is related to the north wind blowing air from the inland, which is more polluted than air from the sea Figure 5a. In particular, no matter which direction the wind is from, NO_2_ is negatively correlated with its concentration, which is related to the small area of Shenzhen (1996 square kilometers) and the high density of cars (more than 3 million). However, the primary source of NO_2_ is cars, so no matter which direction the wind comes from, the concentration of NO2 will decrease Figure 5b.

Air pollutants’ impact on hospitalization in all groups was pronounced after 4–6 days (except PM_10_ female, PM_10_ age 2, PM_2.5_ female, SO_2_ female, and NO_2_ female), and RR began to fall on the 7th day. Chen et al. conducted a relevant study on air pollutants and emergency admission risk in 31 cities of China, which shows that the risk of emergency admission caused by several major air pollutants mainly increased in the lag of 0–2 days [22]. Similar studies conducted by Zhao et al. in Dongguan city, near Shenzhen city, also revealed similar results, with the highest outpatient visits at Lag1 [23]. However, the lag days were too short to detect the whole process in a small number of studies.

Also, after exposure to air pollution, the female was more likely to have severe adverse health effects (in the form of hospitalization). The peak RR of males was between lag4 and lag6, while females’ peak was between lag0 and lag1 in the PM_10_ group of this study. Meanwhile, we observed similar trends in PM_2.5_, SO_2_, NO_2_, and O_3_ groups. The study of Luong et al. on PM_2.5_ and acute lower respiratory infection in children pointed out that the effect of PM_2.5_ on males seemed to be stronger than that on females [24]. However, we have a similar explanation for this result, and the exposure determines the response. In China, women tend to have more outdoor activities than men, which leads to more exposure. Another possibility is that different genders have different air pollutants’ sensitivity levels, which requires additional experiments to prove.

Previous studies on the health effects of air pollutants focused on children [25,26]. Some studies show that increased concentration of PM_10_ is associated with the increase of respiratory hospitalization in all age groups, with the most significant impact on the population between 16 and 59 years old [27]. However, all air pollutants except CO had the most significant impact on the elderly group in our study, the elderly. So we found that the elderly could be the group that needs more attention. A study reported in France that particulate matter also affects the cardiovascular system of the elderly most significantly [28]. The problem of aging is one of China’s biggest social problems and many other parts of the world [29]. We should not ignore the pressure of medical insurance brought by hospitalization and the influence of public health. Therefore, it is necessary and meaningful to carry out more relevant studies on the elderly and take more air quality protection measures.

In other studies on the health effects of various pollutants, most of them apply 10 μg/m^3^ as concentration change into all the pollutants included in the study. However, for different pollutants, 10 μg/m^3^ can be a large unit or a small one. For example, the average concentration of SO_2_ in Shenzhen in 2013 was only 11.84 μg/m^3^. If we used the variation of 10 μg/m^3^ as the model’s parameter, it is easy to get a maximum RR value. For example, a study conducted by Chen et al. in Jinan, China [30] also makes the health effects of multiple pollutants in the same study incomparable. To solve this problem, we set the unit of concentration changes of pollutants to 10% of each pollutant’s annual average value. Under this relatively straightforward design, the comparison of the health effects of various pollutants becomes intuitive and precise. We found that RR of PM_10_ is the biggest one of air pollutants for the population as a whole, and PM_2.5_ is also significantly impacted. We can found that the health threat of inhalable particles is still the major environmental problem in Shenzhen city, which has a high economic level and less-polluting industries.

Inhalable particles have been the focus of research for a long time due to their characteristics, such as staying in the air for a long time, complex chemical composition, attaching pathogenic microorganisms, and acting on different respiratory tract depths particle size. From 1990 to 2016, the overall mortality caused by PM_2.5_ in Iran was on the rise, and researchers believe that air pollution caused a heavy burden on the death rate and years of life lost (YLL) [31]. In Taiwan, a study proposed that inhalable particles can cause nasopharyngeal cancer [32]. A study of inhalable particles from the French offered a novel and practical perspective. The researchers assumed that if all passenger vehicles in Paris met the Euro 5 standard, 148.79 non-accidental deaths would be avoided per year [33]. Automobile exhaust is one of the primary air pollution sources in Shenzhen city. Promoting the use of new energy vehicles may bring significant development to the control of inhalable particles.

## 5. Conclusions

Shenzhen’s air quality is better than in other megacities in China. However, all the pollutants except CO impact the respiratory system’s hospitalization in a short period, and PM_10_ has the most significant impact. Our study in Shenzhen city offers a valuable opportunity for the other similar cities in the world to study air pollution on health effects under the model of high development-low pollution.

Our study also has some limitations. First of all, we collected data from 98 hospitals, and although they all had professional teams, it was challenging to ensure that every hospital had the same standards for diagnosis and hospital admission. Secondly, due to the limited period of collecting data, we could not study the relationship between air pollution exposure and respiratory disease hospitalization over a long period. Thirdly, due to the medical information data release policy in Shenzhen, we can not access the latest data. Although we only obtained data for 2013, we believe that the research still has significance currently. However, we cannot ignore that long-term exposure to air pollution determines that this is a problem. We will also focus on addressing the long-term health effects of air pollutants in future studies. In the end, we carried out this study in Shenzhen city, so the results are only suitable to promote in areas with similar environments, economic conditions, and medical levels.

## Figures and Tables

**Figure 1 ijerph-18-05120-f001:**
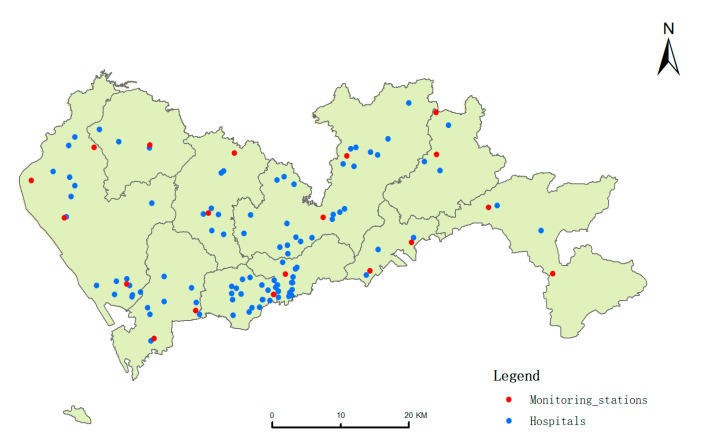
Distribution of 98 hospitals and 19 air quality monitoring stations in Shenzhen.

**Figure 2 ijerph-18-05120-f002:**
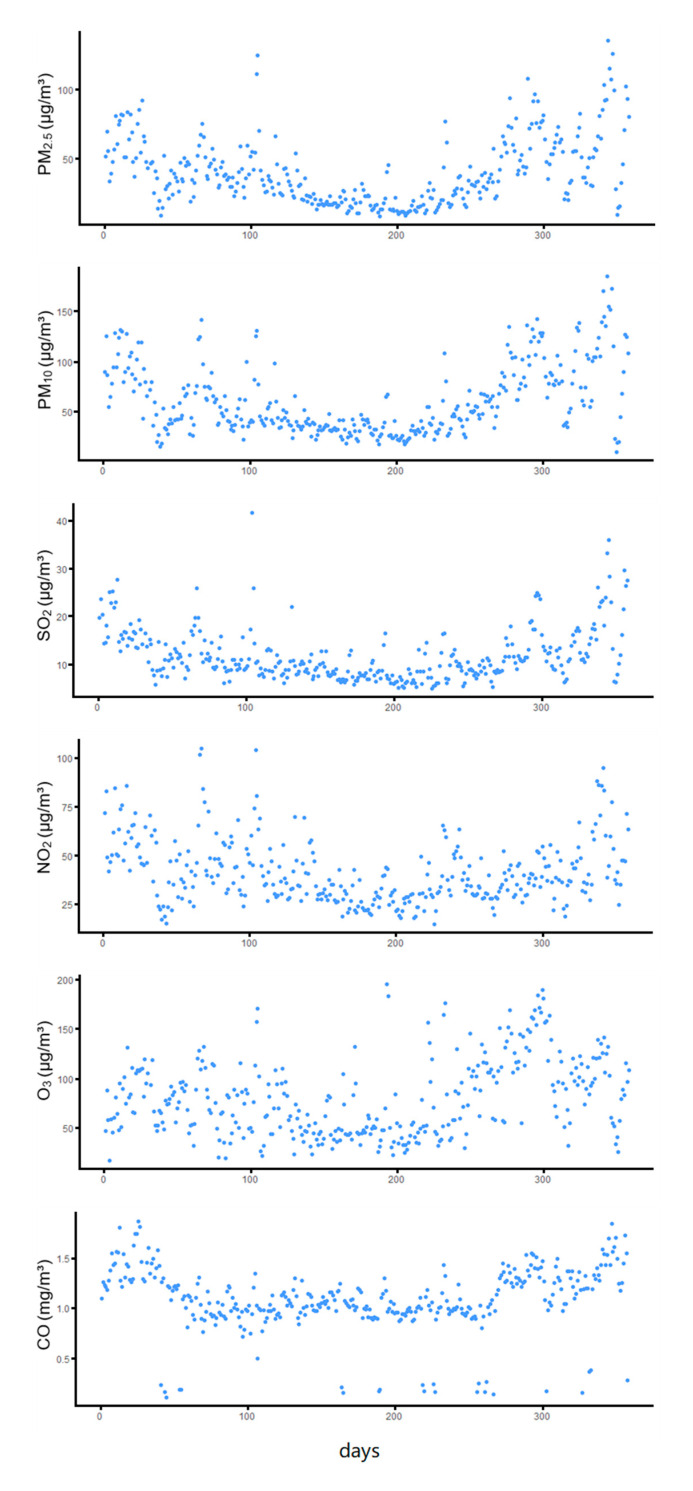
Air pollutant concentration-time scatters diagram.

**Figure 3 ijerph-18-05120-f003:**
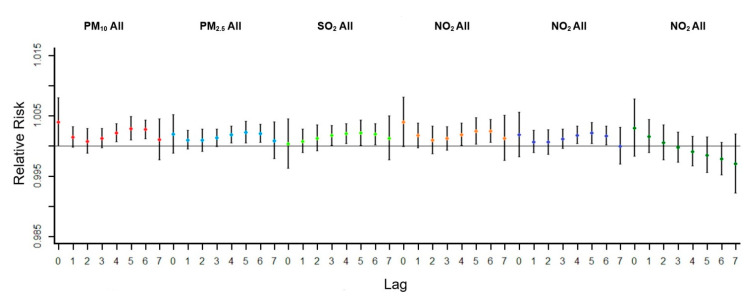
The relative risk of population-wide hospitalization for respiratory disease due to a 10% annual average increase in air pollutant concentration.

**Figure 4 ijerph-18-05120-f004:**
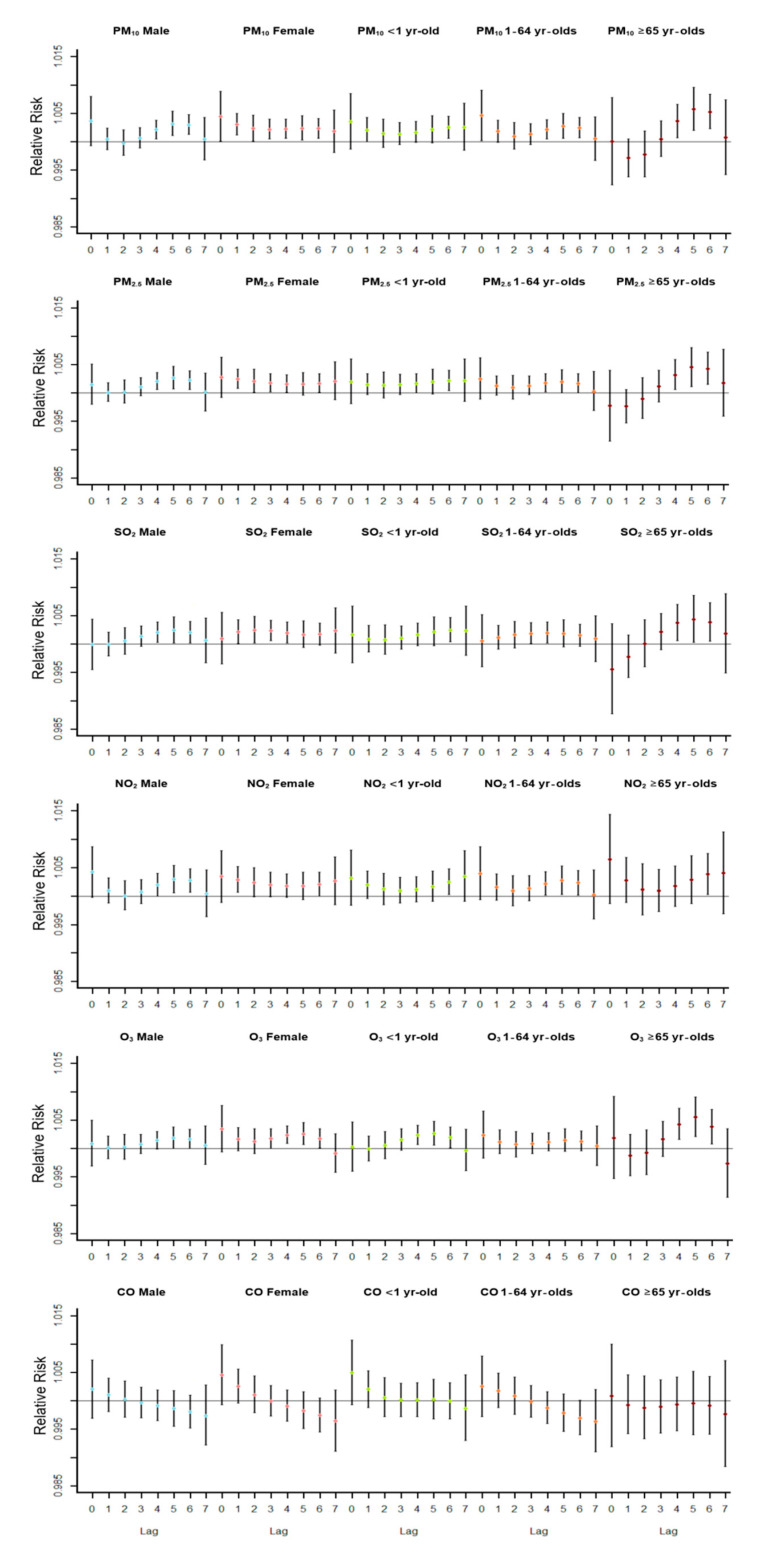
The relative risk of hospitalization for respiratory diseases in different populations was caused by an annual average of air pollutant concentration increased by 10%.

**Figure 5 ijerph-18-05120-f005:**
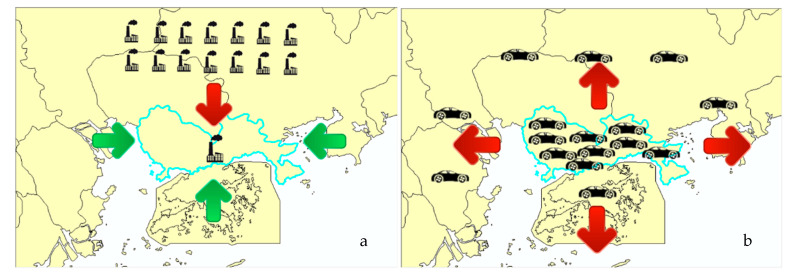
Relationship between wind direction and pollution in Shenzhen. (**a**): Except for NO_2_, the concentration of all pollutants increases with the north wind; (**b**): The concentration of NO_2_ decreases with the wind in all directions.

**Table 1 ijerph-18-05120-t001:** Descriptive statistics of air pollutants, meteorological conditions, and hospitalizations.

	Frequency Distribution			Minimum Value	Maximum Value	Average Value (Standard Deviation)
Items	25	50	75			
air pollutants						
PM_2.5_(μg/m^3^)	21.28	34.46	53.98	8.26	135.81	40.22(24.48)
PM_10_(μg/m^3^)	35.05	49.41	79.75	10.25	184.78	61.31(34.75)
SO_2_(μg/m^3^)	8.10	10.46	14.18	5.03	41.63	11.84(5.48)
NO_2_(μg/m^3^)	29.55	37.54	48.84	14.83	104.81	41.29(16.45)
O_3_(μg/m^3^)	48.25	74.27	107.77	17.32	195.18	80.49(38.53)
CO(mg/m^3^)	0.96	1.08	1.28	0.11	1.86	1.09(0.32)
meteorological conditions						
temperature(°C)	20.00	25.00	28.00	9.00	31.00	23.85(5.02)
humidity(%)	68.00	78.00	87.00	24.00	100.00	75.62(14.69)
The average daily number of hospital admission for respiratory diseases	275.00	310.00	340.00	82.00	417.00	307.92(52.49)
male	168.00	188.00	209.00	46.00	274.00	188.42(33.49)
female	107.00	121.00	134.00	30.00	177.00	119.50(21.86)
<1 year	59.50	79.00	93.00	20.00	125.00	76.58(20.66)
1–64 years old	183.50	204.00	227.00	47.00	310.00	203.32(38.55)
≥65 years old	23.00	27.00	33.00	8.00	56.00	28.01(7.35)

**Table 2 ijerph-18-05120-t002:** Spearman correlation analysis results of air pollutant concentration in Shenzhen city with meteorological factors and wind direction.

	SO_2_	NO_2_	CO	O_3_	PM_10_	PM_2.5_	AQI	Air Temperature	Humidity	East Wind	South Wind	West Wind	North Wind
SO_2_	1.000												
NO_2_	0.745 **	1.000											
CO	0.510 **	0.459 **	1.000										
O_3_	0.583 **	0.317 **	0.359 **	1.000									
PM_10_	0.831 **	0.641 **	0.531 **	0.728 **	1.000								
PM_2.5_	0.815 **	0.667 **	0.576 **	0.684 **	0.947 **	1.000							
AQI	0.824 **	0.767 **	0.593 **	0.677 **	0.904 **	0.929 **	1.000						
air temperature	−0.506 **	−0.447 **	−0.410 **	−0.173 **	−0.433 **	−0.543 **	−0.437 **	1.000					
humidity	−0.630 **	−0.206 **	−0.321 **	−0.662 **	−0.667 **	−0.558 **	−0.499 **	0.265 **	1.000				
East wind	−0.214 **	−0.274 **	−0.059	−0.043	−0.098	−0.097	−0.135 *	−0.012	0.003	1.000			
south wind	−0.259 **	−0.236 **	−0.114 *	−0.145 **	−0.158 **	−0.180 **	−0.206 **	0.157 **	0.123 *	0.404 **	1.000		
west wind	−0.100	−0.128 *	−0.021	−0.057	−0.043	−0.055	−0.079	0.142 **	0.030	−0.037	0.460 **	1.000	
North wind	0.016	−0.112 *	0.065	0.049	0.034	0.036	0.013	−0.108 *	−0.121 *	0.567 **	0.022	0.091	1.000

*: *p* < 0.05, **: *p* < 0.01.

## Data Availability

The weather data of Shenzhen was gotten from the open website http://lishi.tianqi.com/shenzhen/2013 (accessed on 1 June 2018). Data sharing is not applicable to this article.

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
