# Peer review of "The Influence of Air Pollutants and Meteorological Conditions on the Hospitalization for Respiratory Diseases in Shenzhen City, China"

_ijerph, 2021, doi:10.3390/ijerph18105120_

Round 1
Reviewer 1 Report
You can get more conclusions from the work and it has some space errors that are highlighted in green.

Author Response
Point: You can get more conclusions from the work and it has some space errors that are highlighted in green.
Response: You are right. I have revised the Relevant parts. Thank you so much.
Reviewer 2 Report
In general the manuscript needs to be improved to be accepted:
Summary:
It must be reorganized and actually present the relevant findings
Introduction:
It presents too much rhetorical information that does not add greater value to the manuscript, lines 34-37, 45-46, 46-53, 54-55. others need support or greater clarity lines 57-58, 60-61.
The objective described in the introduction differs from the summary, I suggest that they focus on aspects such as exposure time or accumulation, which is what they really propose as novel
Materials and methods:
The data used to make the analysis correspond to 2013, which is something that does not reflect the possible current reality of the phenomenon. They are over 8 years old. What does this year contribute particularly to the authors' analysis?
Line 96 the direction of the wind is not a meteorological factor?
If the authors are consistent with a non-normal distribution (Line 106) of the data, Why propose a linear model? (line 108)
Line 112 How was lag time determined?
Lines 119-121 it is not clear what the authors propose
Line 124 why dummy for the day?
What is the reason for presenting a correlation between weather conditions and pollutants?
Please, improve image quality
Please, Report how many weather stations did they use? and the source of information
Results Results the first paragraph of the results repeats information presented in the table. In addition, the relevance of figure 2? Why not include the summary of the wind speed and direction? Table 2 and the second paragraph also present data duplication. The correlation between PM2.5 and PM10 is not implicit, a collinearity analysis was performed, remember that PM2.5 is part of PM10. The authors considered the half-life of each pollutant when performing the analysis and determining the lag period. What inclusion and exclusion criteria were used to determine the hospitalized population. Why were only age and sex considered as demographic covariates? Was it possible to determine some other aspects such as comorbidities? In the methodology, it is not clear how each monitoring station is associated with each group of hospital data, nor is it clear if what is summarized in the tables and figures corresponds to the average number of stations / hospitals, there is a great gap in how they process the data. in the model and correlations. Finally, the hospital where a patient attends to be treated possibly corresponds to the one closest to his home, however, this does not mean that it is the place where he is most exposed to contamination (school, work, etc.) as the authors control for this variable? Discussion It is necessary to clarify the design aspects in the methodology in order to validate the results and their discussion. Conclusions The conclusions presented do not add much novelty. They do not respond to the proposed objective (s)
Author Response
Authors' Responses to Reviewer's Comments (Reviewer 2)
Comments and Suggestions for Authors:
In general the manuscript needs to be improved to be accepted:
Summary:
It must be reorganized and actually present the relevant findings
Response 1: Thanks to the reviewer for the comment. We have re-written the abstract part of the article according to the suggestion. First, we introduced the research objectives and background of this research and then introduced the research date and primary method of this research, and we summarized the main results and conclusion.
Introduction:
It presents too much rhetorical information that does not add greater value to the manuscript, lines 34-37, 45-46, 46-53, 54-55. others need support or greater clarity lines 57-58, 60-61.
Response 2: Thanks to the reviewer for the comment. We improved the readability and pertinence of the introduction and deleted rhetorical information in lines 34-37, 45-46, 46-53, 54-55. Such as, "While enjoying the advancements of civilization", "However, the hazard of air pollutants is only the tip of the iceberg".
The objective described in the introduction differs from the summary, I suggest that they focus on aspects such as exposure time or accumulation, which is what they really propose as novel.
Response 3: We adopt the reviewer's advice. We have re-written the introduction's objective (lines 57-58, 60-61) to match the abstract part. This study aimed to investigate the effects of air pollutants and meteorological conditions on residents' hospitalization for respiratory diseases in the short term and compare different pollutants' exposure time or accumulation effects.
Materials and methods:
The data used to make the analysis correspond to 2013, which is something that does not reflect the possible current reality of the phenomenon. They are over 8 years old. What does this year contribute particularly to the authors' analysis?
Response 4: To be honest, it is over eight years old since 2013. The situation changed a lot. However, due to the public health information release policy, we did not gain the latest hospitalization data and only got data in 2013.
Line 96 the direction of the wind is not a meteorological factor?
Response 5: We are sorry to make you confused. We add wind direction as a meteorological factor in the description.
If the authors are consistent with a non-normal distribution (Line 106) of the data, Why propose a linear model? (line 108)
Response 6: Please forgive us for the error in the manuscript. We checked our code and model. We actually used the Distributed Lag Non-linear Model (DLNM) package in R instead of a linear model. Furthermore, we modified the model detail described in section 2.4. Methods.
Line 112 How was lag time determined?
Response 7: Thank you very much for your careful reviews. This sentence should be a typographical error and we delete this sentence.
Lines 119-121 it is not clear what the authors propose
Response 8: We designed to discuss the long-term trends and seasonal effects in the original study. However, we did not adopt this part in the final manuscript, and we delete this irrelevant statement.
Line 124 why dummy for the day?
Response 9: In the Distributed Lag Non-linear Model, we added a dummy variable for the day of the week.
What is the reason for presenting a correlation between weather conditions and pollutants?
Response 10: There was a significant negative or positive correlation between the wind of different directions and pollutants. Details can be found in section 3.1 and the Discussion.
Please, improve image quality
Response 11: Thank you for the suggestion.We have redrawn Figure 1 and replaced other figures with clear images.
Please, Report how many weather stations did they use? and the source of information.
Response 12: We gathered meteorological data (temperature, humidity, and wind direction) of Shenzhen from the open website http://lishi.tianqi.com/shenzhen/2013. Provided data is the average weather conditions of the entire Shenzhen, we don't know that these data are obtained from several sites.
Results Results the first paragraph of the results repeats information presented in the table. In addition, the relevance of figure 2? Why not include the summary of the wind speed and direction?
Response 13: Thanks to the reviewer for the comment. We summarize the main findings from Table 1 in the first paragraph to help readers read and understand Table1. We include the summary of the wind speed and direction in table2. However, it is not easy to add wind speed and direction in figure 2.
Table 2 and the second paragraph also present data duplication. The correlation between PM2.5 and PM10 is not implicit, a collinearity analysis was performed, remember that PM2.5 is part of PM10. The authors considered the half-life of each pollutant when performing the analysis and determining the lag period.
Response 14: Thanks to the reviewer for the comment. We summarize the main findings from Table 2 to help readers read and understand. We agree that PM2.5 comprises a portion of PM10. Nevertheless, PM2.5 and PM10 have a different impact on residents' hospitalization for respiratory diseases. PM2.5 has more impact on Lower respiratory diseases.
What inclusion and exclusion criteria were used to determine the hospitalized population. Why were only age and sex considered as demographic covariates? Was it possible to determine some other aspects such as comorbidities?
Response 15: Thanks to the reviewer for the comment. We discussed age and sex group in the hospitalized data. We agree that it's interesting to discuss some other aspects, such as comorbidities. Limited to current work, we will consider it in future research.
In the methodology, it is not clear how each monitoring station is associated with each group of hospital data, nor is it clear if what is summarized in the tables and figures corresponds to the average number of stations / hospitals, there is a great gap in how they process the data.
Response 16: we gathered daily average weather conditions meteorological data (temperature, humidity, and wind direction) of Shenzhen. We don't know that these data are obtained from several sites. Moreover, it can not match the location of air quality monitoring stations and hospitals.
in the model and correlations. Finally, the hospital where a patient attends to be treated possibly corresponds to the one closest to his home, however, this does not mean that it is the place where he is most exposed to contamination (school, work, etc.) as the authors control for this variable?
Response 17: Thanks to the reviewer for the comment. The average daily concentration of each air pollutant as the exposure concentration was calculated based on the kriging interpolation method in each 1km grid. We geocode the residential location to match exposure concentration in each 1km grid.
Discussion It is necessary to clarify the design aspects in the methodology in order to validate the results and their Discussion.
Response 18: Thanks to the reviewer for the comment. We have improved the 2.4 Methods section. We hope it will be clarified when you read in results and Discussion.
Conclusions The conclusions presented do not add much novelty. They do not respond to the proposed objective (s)
Response 19: Thank you for the suggestion. Some repetitions in the Discussion had been moved to the conclusion. Moreover, we rewrote the conclusion. Shenzhen's air quality is better than other megacities in China. Howerver, all the pollutants except CO impact the respiratory system's hospitalization in a short period, and PM10 has the most significant impact. Our study in Shenzhen city offers a valuable opportunity for the other similar cities in the world to study air pollution on health effects under the model of high development-low pollution.
Reviewer 3 Report
The paper is interesting, but some amendments are necessary.
The English language is not perfect, making some parts (e.g. chapter 3.3) difficult to understand.
All the acronyms must be explained at their first use in the paper.
The statistical methods should be better presented.
The Discussion contains some repetitions and could partly be moved to Conclusion, that, as is, seems too short.
A good global revision will surely improve the paper, making it suitable for publication. The paper is interesting, but needs a deep revision.
Author Response
Authors' Responses to Reviewer's Comments (Reviewer 3)
Comments and Suggestions for Authors:
The paper is interesting, but some amendments are necessary.
The English language is not perfect, making some parts (e.g. chapter 3.3) difficult to understand.
Response 1: Thank you very much for your careful reviews on our manuscript. We have invited a native speaker to review the manuscript content to improve readability sufficiently.
All the acronyms must be explained at their first use in the paper.
Response 2: We adopt reviewer's advise. We explained all the acronyms at their first use in the manuscript. Such as, Particulate Matter (PM10, PM2.5) 、Nitrogen dioxide (NO2)、Ozone (O3)、Sulphur dioxide (SO2)、Carbon monoxide (CO) and so on.
The statistical methods should be better presented.
Response 3: Thanks to the reviewer for the comment. We have improved the 2.4 Methods section. We checked our code and model. We actually used the Distributed Lag Non-linear Model (DLNM) package in R instead of a linear model. Furthermore, we modified the model detail described in section 2.4. Methods.
The Discussion contains some repetitions and could partly be moved to conclusion, that, as is, seems too short.
Response 4: Thank you for the suggestion. Some repetitions in the Discussion had been moved to the conclusion. Moreover, we rewrote the conclusion.
A good global revision will surely improve the paper, making it suitable for publication. The paper is interesting, but needs a deep revision.
Response 5: Thank you for the suggestion. Shenzhen's air quality is better than other megacities in China. Howerver, all the pollutants except CO impact the respiratory system's hospitalization in a short period, and PM10 has the most significant impact. Our study in Shenzhen city offers a valuable opportunity for the other similar cities in the world to study air pollution on health effects under the model of high development-low pollution. And we point limitations in our study.

Round 2
Reviewer 2 Report
Thanks to the authors for improving your work, however you still have conceptual problems as well as data control.
- The authors acknowledge that the current air quality situation is substantially different between 2013 and the current one. This reflects that it is not correct to generate a publication that does not reflect the current state of the problem.
- Although the authors respond to my questions, they do not correct the lack of conceptual information
- The interpolation gives a high resolution of air quality, but this does not mean that the information is reliable, remember that they themselves talk about atmospheric transport from surrounding areas in the discussion. That makes me think that they do have notions of the implications of pollutant dispersion, and that an interpolation by itself is not a reliable tool to determine an air quality parameter. the method used ignores the physical and chemical processes of atmospheric transport.
- Carrying out kriging does not solve my question about the control of patients admitted to X hospital and the area and where they are exposed. the city is densely populated and according to estimates made in figure one is only 40 km long. so a person can spend time of exposure more than 20 km from where it is entered. the selection of the hospital is based on the proximity to your residence, which may differ from your place of work or study. The authors are not clear on how to control the mobility of the population, so the correlation of air quality with hospitals has a great limitation.
- The fact that the authors do not have quality control and certainty of the meteorological information is a factor that makes part of the investigation less valid. From the statistical point of view, it is not clear how many times the same meteorological data is assigned to the 19 air quality stations, so a correlation of the same value (meteorological parameter) with a large range of data variation (weather stations) air quality) can lead to coefficient failures.
Taking into account these flaws in the design of the document, I continue to argue that it is not feasible to accept this manuscript for publication.
Reviewer 3 Report
The paper was improved, but some issues should still be addressed.
In all the text: pay attention to the subscripts
line 21: "revealed" - better "investigated"
line 41: PM2.5 has already been defined
line 53: better "Therefore, it is not reliable to determine the effects on health..."
line 58: better "...to study the effects of air pollution on health..."
line 73: "is" - better "lasts"
line 113: "3.5.1" ? Is it a version number?
line 114: "to construct to examine" ?
line 119: "model" - better "equation"
line 121: add "where" at the beginning
line 127: better "...effects on health..."
line 129: wide ?
line 185-208: please clarify "lay" and "lag"
line 206-208: please rewrite to clarify
line 228: "increasing" - better "increases" ; "decreasing" - better "decreases"
line 278: please explain "YLL" meaning
line 286: "...is better than in other..."
Ref. 19: please verify format
A final proof-reading is recommended
